# Management of Anterocapitis and Anterocollis: A Novel Ultrasound Guided Approach Combined with Electromyography for Botulinum Toxin Injection of Longus Colli and Longus Capitis

**DOI:** 10.3390/toxins12100626

**Published:** 2020-09-30

**Authors:** Michael Farrell, Barbara I. Karp, Panagiotis Kassavetis, William Berrigan, Simge Yonter, Debra Ehrlich, Katharine E. Alter

**Affiliations:** 1MedStar/Georgetown University National Rehabilitation Hospital, Washington, DC 20010, USA; Michael.E.Farrell@medstar.net; 2National Institutes of Neurological Disorders and Stroke, Bethesda, MD 20892 USA; Barbara.Karp@nih.gov (B.I.K.); panagiotis.kassavetis@nih.gov (P.K.); Debra.Ehrlich@nih.gov (D.E.); 3Emory School of Medicine, Emory University, Atlanta, GA 30322, USA; william.alvin.berrigan@emory.edu; 4Rehabilitation Medicine, Clinical Center, National Institutes of Health, Bethesda, MD 20892-1604, USA; simge.yonter@nih.gov; 5Functional and Applied Biomechanics Section, Rehabilitation Medicine, Clinical Center, National Institutes of Health, Bethesda, MD 20892-1604, USA

**Keywords:** cervical dystonia, botulinum toxins, anterocaput, anterocollis, injection technique, longus colli, longus capitis, ultrasound guidance, head and neck, chemodenervation

## Abstract

Chemodenervation of cervical musculature using botulinum neurotoxin (BoNT) is established as the gold standard or treatment of choice for management of Cervical Dystonia (CD). The success of BoNT procedures is measured by improved symptomology while minimizing side effects and is dependent upon many factors including: clinical pattern recognition, identifying contributory muscles, BoNT dosage, and locating and safely injecting target muscles. In patients with CD, treatment of anterocollis (forward flexion of the neck) and anterocaput (anterocapitis) (forward flexion of the head) are inarguably challenging. The longus Colli (LoCol) and longus capitis (LoCap) muscles, two deep cervical spine and head flexor muscles, frequently contribute to these patterns. Localizing and safely injecting these muscles is particularly challenging owing to their deep location and the complex regional anatomy which includes critical neurovascular and other structures. Ultrasound (US) guidance provides direct visualization of the LoCol, LoCap, other cervical muscles and adjacent structures reducing the risks and side effects while improving the clinical outcome of BoNT for these conditions. The addition of electromyography (EMG) provides confirmation of muscle activity within the target muscle. Within this manuscript, we present a technical description of a novel US guided approach (combined with EMG) for BoNT injection into the LoCol and LoCap muscles for the management of anterocollis and anterocaput in patients with CD.

## 1. Introduction

Dystonia is a neurological movement disorder in which sustained or repetitive muscle contractions result in twisting and repetitive movements or abnormal fixed posture [1]. Cervical Dystonia (CD) is the most common focal dystonia with a reported prevalence ranging up to 4100 cases per million with an incidence of 8–12 cases per million person-years [2].

The symptoms of CD include abnormal neck postures, involuntary head and neck movements, tremor, neck pain, and even involved muscular hypertrophy [3]. CD can be associated with similar dystonic symptoms in nearby muscles, such as those in the shoulder, upper back and/or face. These symptoms often have a debilitating impact on an individual’s function and quality of life.

Effective management of CD requires a thoughtful physical examination of the head and neck musculature, observation of involuntary movements and postures, and patient report of muscle pulling, tightness and pain. This evaluation helps to guide muscle selection for targeted treatment. Predominant movement patterns are used to subtype CD into torticollis (neck/head turning), laterocollis (tilting), anterocollis (Figure 1a), anterocaput (Figure 1b) (neck or head flexion, respectively), or retrocollis/retrocaput (neck/head extension) and or lateral shift (combination of laterocollis to one side and laterocaput to the opposite side) [4,5]. Patients may also have a combination of these postures [6] (Figure 1c, Appendix A). Predominant anterocollis occurs in 25% of those with cervical dystonia [7].

The gold standard of treatment for focal dystonias, including CD, is botulinum toxin (BoNT). The efficacy of BoNT treatment depends on a number of factors including proper muscle selection based on patient presentation. The more complex the cervical dystonia, the more difficult the muscle selection [8].

The seven (or eight) known serotypes of botulinum toxin are neurotoxins produced by the Clostridium botulinum bacterium, all of which act at presynaptic neuromuscular (NMJ) and neuro-glandular junctions to block the release of neurotransmitters (acetylcholine and others) from presynaptic vesicles [9,10]. When injected in muscle, BoNTs decrease the release of acetylcholine at the NMJ, producing graded/dose-dependent, reversible, denervation weakness [11]. The mechanism of action in relieving dystonia symptoms is not solely due to its weakening effects on muscle. BoNTs also have effects on various central and peripheral pathways implicated in the pathogenesis of dystonia. The efficacy of BoNT is well-documented in the literature for the treatment of muscle overactivity syndromes, including CD with satisfactory symptom relief in approximately 80%–85% of cases [12,13]. Both the efficacy and safety of BoNT therapy is largely dependent on delivery of BoNT into the affected muscles contributing to dystonic postures/movements while avoiding untargeted muscles and other structures. Accurately placing BoNT within a muscle while avoiding other structures relies heavily on accessibility of the muscle which is directly influenced by regional anatomy. To increase the accuracy and safety of injections, clinicians utilize a number of localization techniques including anatomic landmarks, palpation, range of motion, electromyography (EMG), electrical stimulation (E-Stim) and imaging such as fluoroscopy, CT guidance, and ultrasound [3,14,15].

BoNT injections to address anterocaput/anterocollis by targeting the LoCap and LoCol muscles are challenging due to anatomical and technical factors created by their location deep in the anterior neck. A thorough understanding of regional anatomy is key for clinicians performing BoNT injections into these and other muscles targeted when treating CD.

## 2. Results

### 2.1. A Novel Combined US Guided-EMG Approach to BoNT Injections into the Longus Colli and Capitis Muscles: Review of Relevant Regional Anatomy

The LoCol and LoCap lie in the deep, anterior cervical compartment, lying in a groove created by the anterior tubercle of the transverse processes (TvP) and vertebral body. These muscles are deep in the prevertebral fascia as well as structures in the anterior neck including other muscles including those involved in deglutition, the esophagus, neurovascular structures, and the thyroid gland [16]. Clinicians who inject these and other muscles when treating patients with CD must be aware of and avoid the many critical structures which often lie on the path to or adjacent to the target muscle or muscles. (Figure 2a,b).

Supporting functional anatomic studies in unimpaired individuals suggest that the deeper cervical flexor muscles, including the LoCol and LoCap play a critical role in head and neck flexion at the craniocervical junctions [17]. The four slips of the LoCap originate on the anterior tubercles of the C3–C6 transverse processes (TvP) and run superiorly and medially to insert on the basilar portion of the occipital bone. When contracting bilaterally, the LoCap acts to flex the head and the proximal portion of the cervical spine. Acting unilaterally, the LoCap contributes to ipsilateral head rotation.

The LoCol originates on the anterior tubercle of the TvP of T3-C5, inserts on the C1 vertebra (Atlas) and is divided into superior, intermediate and inferior portions. Its primary action is flexion of cervical spine or neck. The inferior oblique portion of the LoCol may contribute weakly to ipsilateral flexion and contralateral rotation of the neck. The LoCap and LoCol overlap from C1–C6 and therefore both act on this portion of the cervical spine [16].

When performing BoNT injections into the LoCap/LoCol, avoiding superficial structures in the path of the needle to the target muscles creates extreme technical challenges (Figure 2a,b). While commonly utilized non-imaging guidance methods for BoNT injections (EMG, E-stim, and surface anatomy delineated in reference guides) may be adequate for more superficial muscles, these techniques provide no information about muscle depth or the location of structures to be avoided in an individual patient. In anterocaput/anterocollis, the targeting challenges associated with injection of the deep flexors has led many clinicians to limit injections to the more easily accessible superficial flexor muscles, the sternocleidomastoids (SCMs), anterior scalenes (ASs). The ASs attach to to the anterior tubercle of the Tvp of cervical spine and therefore cannot flex the head. The SCM attaches to the head, posterior to the mastoid and therefore flexes the neck but extends the head. Therefore, injecting only these superficial flexor muscles may provide only limited or no benefit for those patients whose dystonic movements or postures are caused by LoCap or LoCol over-activity [6].

Because of technical challenges and potential adverse events associated with BoNT injections guided by non-imaging techniques, some clinicians have turned to imaging-based guidance as a potentially safer and more reliable localization alternative, especially for deep muscles, such as LoCap and LoCol [6,15]). The development of portable ultrasound (US) equipment makes use of 2-D US particularly appealing.

Allison and Odderson (2016) detailed the technical challenges of BoNT injections into LoCap/LoCo in a case report, where they described a technique for combined US- and EMG- guided lateral and medial injection approaches. This technique is performed at the level of the cricothyroid cartilage using the AS muscle and carotid artery (CA) as anatomic landmarks. The authors advocated the combination of EMG and US guidance as EMG provides an added benefit, e.g., confirmation of activity in the target muscle. The authors also indicated drawbacks of each approach. The use of the lateral approach as described, is often limited by an anatomic variation where the CA is in a more lateral position, blocking access to the muscles of interest. Utilizing the medial approach, necessitates the needle be inserted into and then through the ipsilateral thyroid lobe on its way to the LoCol.

Herein, we present for the first time an alternative, combined US-EMG guidance technique for injection which provides a reliable, and perhaps safer approach, to the LoCap and LoCol. This lateral approach utilizes the C6 and C5 vertebral anterior tubercles of the TvPs as well-defined anatomic landmarks for identifying the adjacent target muscles, the carotid artery, jugular veins, brachial plexus and nerve branches (Appendix A). In the majority of patients this technique allows the needle to be safely inserted into the target muscles while avoiding vessels and nerves and obviating the need to puncture the thyroid gland. A similar approach has been described for ultrasound-guided stellate ganglion block [18,19].

#### 2.1.1. Description of US Guided Localization and Procedure Technique for Injection of the Longus Colli and Capitis Muscles

##### Localization of the LoCol and LoCap with B-Mode and Color Doppler US

The patient is positioned seated on an adjustable examination chair, with the back of the chair sloped approximately 45 degrees backward and with the head resting on a headrest. Prior to injection, the US anatomy of the individual’s neck is thoroughly explored using both B-mode imaging in short and long axes and key structures identified. Scanning is performed with the neck in neutral rotation followed by maximal contralateral rotation, i.e., with the head turned away from the side to be injected. This is to determine if one position or the other provides the safest path to the target. Assessing the patient in both neutral and contralateral rotation is imperative as the optimal position may vary due to anatomic differences or rearrangements associated with dystonia (Figure 2a). An anatomic scan is also required to determine the depth of the target muscle/s so as to select the appropriate length of the injecting monopolar needle electrode.

It is helpful to start each US examination with the transducer initially placed over easily identified superficial structures in the neck, such as over the anterolateral neck whereby the sternocleidomastoid muscle, carotid artery and jugular vein are easily visualized [20]. From this “home base” position, the entire region of interest is then scanned to identify the relevant anatomy including the muscles of interest, individual anatomic variants and other structures (e.g., blood vessels, nerves, masses, lymph nodes), especially those to be avoided. Color Doppler is useful during anatomic scanning to identify blood vessels (Appendix A). The transducer is then moved gradually laterally or posteriorly to visualize structures in the interscalene triangle, including the anterior and middle scalene muscles, the cervical nerve roots, trunks of the brachial plexus and the cervical spine transverse processes.

The cervical vertebral TvPs can be identified by the unique sonoacoustic appearance of their respective anterior and posterior tubercles. When imaged with US in short axis, the anterior and posterior tubercles of C5 are of similar size and the neural foramina opening is relatively narrow. C6, in contrast, has a larger anterior tubercle compared to the posterior tubercle and a relative deep neural foramina opening, whereas C7 has a posterior but no anterior tubercle, thus having the appearance of a chair [21]. Anatomically, C6 is approximately 2 cm below the level of the thyroid cartilage [22].

After the anatomic scan is completed the transducer is returned first to the short axis anterior neck “home-base,” position and then to the inter-scalene region. Once in the inter-scalene position and the vertebral TvPs are visualized. While remaining in short axis, the transducer is moved cranially and caudally to identify the C5 and C6 TvPs and to determine which level provides the optimal path to both the LoCap and LoCol muscles. When scanning, moving the transducer caudally and laterally from “home base” to below the level of the cricoid cartilage will bring the anterior tubercle of the C6 TvPs into view. In many, but not all, patients, the C6 level may provide the best path to the target.

Using this view, the LoCol and LoCap can then be identified lying anterior to the anterior tubercles of the cervical vertebra (Figure 2b). The LoCap lies anterior and slightly lateral to the LoCol muscle at this level. The two muscles are separated by a hyperechoic fascia which is easily identified (Figure 2b). With this view, it can be seen that the carotid and jugular vessels are more anterior and safely away from the planned trajectory of the needle to the muscles (Figure 2a,b). It can also be noted that the vertebral artery is posterior to the anterior tubercle, which helps to protect it from accidental puncture. Color Doppler imaging can be utilized to confirm location of critical blood vessels to avoid. The cervical nerve roots and trunks of the brachial plexus are also more posterior and can thus be circumvented. The transducer position should be adjusted to place the target muscles in the center of display screen.

##### BoNT Injection Procedure

Many practitioners (including the authors) combine EMG with US for BoNT injections using a monopolar EMG/chemodenervation needle connected to an EMG. While US guidance confirms the anatomic location of the needle, the concurrent use of EMG permits muscle activity to be assessed during procedure. Excessive muscle activity at rest can confirm its involvement in the patient’s CD. In our center, combined EMG and US guidance is used for all cervical muscle injections including injections of the LoCap and LoCol muscles.

Prior to injection, the skin is cleaned and aseptic technique is utilized, including covering the US probe with a transducer cover. Prior to beginning the procedure, the injecting monopolar needle electrode is connected to a portable EMG machine or to an electrodiagnostic instrument with the EMG setting enabled. US-guided injection procedures can be performed either “in-plane,” where the needle inserted along the long axis of the transducer (Figure 3a), or “out-of-plane” (Figure 3b) with the needle inserted across the short axis of the transducer (Appendix A). The sonographer/injector should be familiar with the advantages and limitations of each of these techniques [23]. In some patients, an out-of-plane approach may provide the most direct path to these deep cervical target muscles (Figure 2b).

Once the LoCap and LoCol target muscles and safe path are identified, the needle is inserted through the skin and the EMG machine is switched on. The needle is then advanced under continuous US visualization towards the target muscle(s) (Appendix A). Using a freehand technique, the hand holding the transducer is stabilized against the patient’s neck and the “freehand” is used to insert the needle so that the needle trajectory can be continuously adjusted to pass safely adjacent to, but not through, neurovascular structures (Figure 2). When utilizing an “out-of-plane” approach, the authors recommend inserting the needle at a steep angle (relative to the transducer), Figure 3b rather than at a shallow angle (Figure 3c). This technique may reduce the risk of advancing the needles tip beyond the US beam and into untargeted structures. Given this risk associated with out-of-plane needle insertions it is the authors’ opinion that using a steep angle of needle insertion is more desirable than a shallow angle of insertion, regardless of which muscle is injected.

Entry of the needle through the prevertebral fascia and into LoCap muscle can be visually observed with US, the presence or absence of muscle activity confirmed and then the BoNT is injected into the muscle. The needle can then be further advanced into the LoCol muscle and the above process repeated for its injection, if required.

## 3. Discussion

Many experienced BoNT practitioners consider anterocaput/anterocollis the most difficult pattern of CD to successfully manage with BoNT. This difficulty largely stems from the number of muscles that can contribute to these head or neck postures and their locations, both superficial (SCM, AS) and deep (LoCap, LoCol). In all patients with anterocaput and in some patients with anterocollis, chemodenervation of the LoCap or LoCol may be required to improve symptomology. However, due to complex regional anatomy, BoNT injections in these muscles without direct visualization, such as US-guidance, is very challenging and risky. In this paper, we describe a novel approach for visualizing a safe trajectory for injecting the LoCap and LoCol utilizing US guidance. This methodology may enhance efficacy of chemodenervation for CD by enabling injection of involved deep cervical muscles and reduce the risk of adverse events associated with their injection.

The unique aspects of our approach compared to previously described approaches include (1) utilizing the anterior tubercles of the C5 and C6 cervical vertebrae as landmarks whereby the LoCap and LoCol are easily identified and, (2) evaluating the position of the patient’s neck in neutral and maximal contralateral rotation to determine which position creates the best path to the target by placing the carotid artery and sheath anteriorly relative to the muscles and the anterior tubercle. A prior study reported that maximum contralateral rotation of the neck for stellate ganglion blocks created the greatest distance between the carotid artery and the path of the needle to the target [18]. When utilizing this technique for BoNT injections in the LoCol/LoCap scanning in both neutral and contralateral neck rotation will determine which neck position provides the best access to the target muscles and greatest distance from the carotid artery.

A previous report described an approach that was medial or lateral relative to the carotid sheath when injecting the LoCol muscle [6], with the patient’s neck positioned in midline. That medial approach has the disadvantage of requiring that the needle be inserted into and through the thyroid gland, with the attendant risk of puncturing the inferior thyroid artery. Their lateral approach has the disadvantage that the window between the anterior tubercle, muscles and the carotid artery that the needle must pass through to get to the deep muscles is quite narrow when the head is positioned midline. An additional limitation of both of these approaches is that they only permit injection of the most distal portion of LoCap, so that anterocaput may only be incompletely treated.

In contrast, our proposed lateral approach avoids traversing the thyroid gland and also maximizes the anatomical distance between the needle and the carotid sheath, minimizing the risk of accidental puncture of this important vascular structure. A study by Park et al. investigated a similar technique for increasing the safety of US-guided stellate ganglion blocks; they found that the distance between the C6 anterior tubercle and the carotid artery was maximized with complete contralateral rotation of the head [18].

We typically perform these injections at the C6 level where there is overlap between the LoCap and LoCol so that both muscles can be targeted with a single puncture. This same approach can be used at the C5 level if it provides a clearer path to the muscles in the individual patient. At either level, the needle trajectory provides access to both muscles, with the needle first entering the LoCap from which it can be advanced into the LoCol. Injection below C6 is less desirable as below that level the LoCap is thinner and the LoCol lies deeper as it moves obliquely to insert on transverse processes in the upper thoracic region.

In our experience, this technique provides reliable, consistent access to two difficult to approach muscles in most patients and is useful when injecting the deep head/neck flexor muscles. However, there are potential limitations to this technique and cautions for the clinicians who perform these procedures;
In some patients the described approach does not provide a safe window through which to insert the needle into the LoCap or LoCol. This may be due to anatomical variations, rearrangements, involuntary movements or limited cervical range-of-motion. When this situation is encountered, and a safe path to the target muscles cannot bet identified, we do not insert the needle or perform injections because of the risk of damage to critical vessels or nerves. It is important to acknowledge that this technique reduces, but does not completely abolish, the risks of deep cervical muscle injection.Many clinicians prefer utilizing an “in-plane” approach for US guided injections. However, in some patients an “out-of-plane” approach may provide more direct access to the LoCol, LoCap and other muscles. Therefore, clinicians must be skilled at both approaches.This is an advanced US/chemodenervation technique that must be conducted by clinicians who are skilled in performing US-guided injections to minimize risk to the patient.Prior to performing such an injection, procedural consent should be obtained so that the patient is informed of the potential procedural risks and as well as the potential benefits.Because of proximity to muscles involved in deglutition and other cervical muscles, injection of these deep muscles may be more likely to cause swallowing difficulties, speech problems, or more rarely, breathing problems than injections of more typical superficial muscles in patients with CD.

## 4. Conclusions

Clinicians who treat patients with CD should consider learning this technique to potentially improve the outcome when treating patients with anterocaput and anterocollis by enabling access to deep cervical muscles for BoNT injection, and decreasing the risk associated with injection of these muscles. The described technique provides clear and key information on patient positioning, US probe placement, US anatomical landmarks and needle trajectory to improve the safety of injections into the LoCap and LoCol muscles.

## 5. Materials and Methods

This study was approved by the NIH Institutional Review Board and written informed consents were obtained from all patients (No. 85N0195; Date: 24 April 2020).

The procedural technique for novel technique described in this manuscript is provided in the BoNT Injection Procedure section, above.

Room requirements: dimmable overhead lighting and or a procedure lamp, an electric adjustable height and back exam table and multiple grounded electrical outlets.

### 5.1. Procedural Equipment

An ultrasound machine with a musculoskeletal preset, depth, gain and focal zone adjustment. Needle enhancement software (beam steering) is useful for in-plane procedures. A foot switch for hands-free capture of procedural images/cine-loops.

Transducers: at minimum a linear transducer with a high frequency range of 12–18 MHz and a low frequency range of 5–7 MHz.

US Supplies: ultrasound gel, a transducer cover with sterile gel, topical skin preparation, gloves, sterile gauze, tissues and damp towels to remove gel, and band aids.

EMG machine: either portable EMG, EMG-electrical stimulation unit or diagnostic EMG machine.

EMG supplies: surface electrodes, monopolar Teflon coated injection needles typically 25–37 mm, 26–27 gauge, batteries for portable units.

### 5.2. Procedural Ergonomics

The patient and the injector should be positioned in line with the US display screen and machine controls. This allows the injector to continuously observe the patient, the needle/syringe and display screen without turning their head. This is imperative during all US guided procedures to reduce the risk of a change in position of the transducer or needle relate to the patient and structure to be injected.

The exam table height should be adjusted to position the patient’s head/neck upper torso at a comfortable height for the injector to either sit on an exam stool or stand so as to avoid reaching above shoulder height.

An assistant for the injector while not required, is useful to help with operation of the EMG machine.

## Figures and Tables

**Figure 1 toxins-12-00626-f001:**
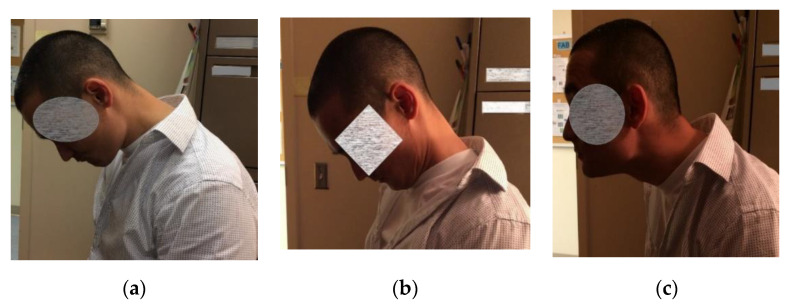
Clinical patterns of cervical dystonia. (**a**) anterocollis; (**b**) anterocapitis; (**c**) combined pattern of anterocollis and retrocapitis.

**Figure 2 toxins-12-00626-f002:**
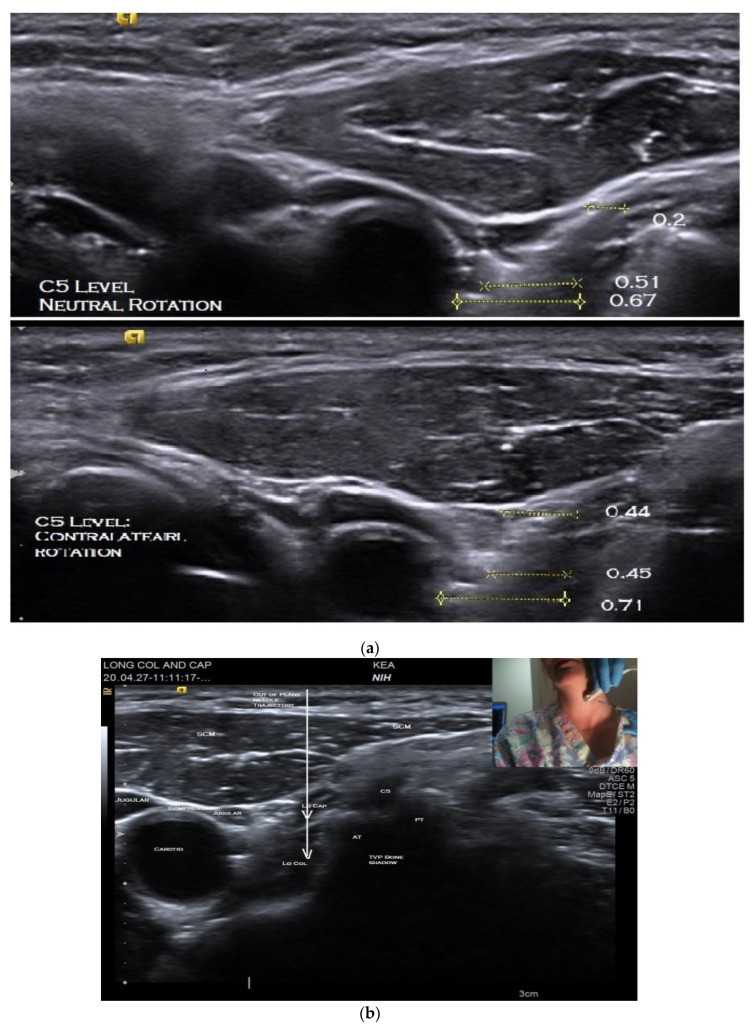
Short axis B-mode ultrasound images of longus capitis and colli muscles. (**a**) C5 level, in neutral position (top) and maximum contralateral rotation (bottom); (**b**) C5 level, with position of needle insertion and trajectory for out-of-plane injection and transducer position (inset).

**Figure 3 toxins-12-00626-f003:**
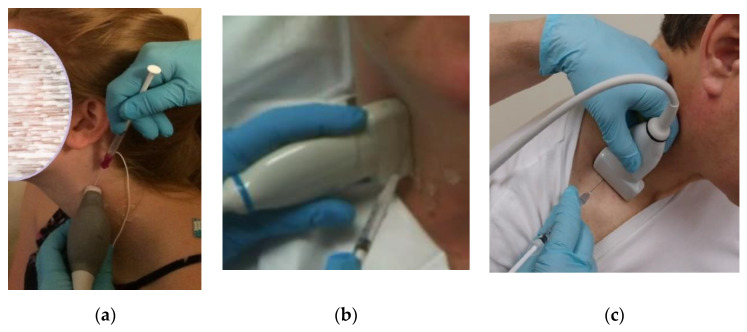
Examples of ultrasound guided needle insertion techniques (**a**) in-plane needle insertion technique (sternocleidomastoid muscle); (**b**) out-of-plane technique with steep angle of needle insertion (longus Capitis/Colli muscles); (**c**) out-of-plane with shallow (less desirable) needle angle (Scalenes).

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
