# Peer review of "Management of Anterocapitis and Anterocollis: A Novel Ultrasound Guided Approach Combined with Electromyography for Botulinum Toxin Injection of Longus Colli and Longus Capitis"

_toxins, 2020, doi:10.3390/toxins12100626_

Round 1

Reviewer 1 Report

This is a generally interesting and well-written manuscript. The topic is certainly of clinical relevance for diagnosing and treating patients with cervical dystonia with predominant antecollis/antecaput. The methodology is well described (with some additional suggestions for improvement, see below), and the authors adequately discuss the potential advantages and limitations of this BoNT injection technique.

Although the description of this novel injection technique will probably be useful for clinicians treating patients with antecollis/antecaput with BoNT, a (small) clinical study or at least a case series of patients who have been treated using this technique with documented benefit would be required to actually demonstrate its potential advantage as compared to existing injection techniques.

Specific comments:

  1. P.2, line 45 and fig. legend (“anterocapitis” and “retrocapitis”): it should be “laterocaput” instead of “laterocapitis”; further, the concept of latercollis/laterocaput and torticollis/torticaput (“ColCap”) should also be mentioned respectively briefly explained (Finsterer et al., 2015; Jost et al., 2020).
  2. A small figure depicting the anatomy of LoCol and LoCap and, if possible, the approach for injection described in this manuscript in contrast to the existing approaches would be helpful for illustration, if there is enough space. The videos provided as supplementary material are indeed helpful.
  3. P. 3, l. 102: “… do not flex the head or proximal cervical spine…”: I doubt if this statement can be accepted in its absoluteness; although LoCol and LoCap certainly have an important role in flexing the head and cervical spine, SCMs and ASs are nevertheless able to contribute to this movement.
  4. P. 8, l. 276-340 is completely redundant/identical to p. 6, l. 213-275 (by mistake?).

Author Response

  1. Thank you to reviewer 1 for suggesting a small clinical study reporting on the outcomes of injections guided with the technique described in this manuscript.. We are in the process of collecting data for a follow up study to specifically address the safety and efficacy of the technique.
  2. Line 45: Corrected as suggested by reviewer. The reviewer also suggested a description of the concept of laterocollis, laterocaput, torticollis/torticput. A brief description of these clinical patterns is provided in lines 42-45. Since the focus of this article is on anterocollis/anterocaput we did not expand on this brief description in the manuscript
  3. Due to the number of figures in this manuscript and limited access to an illustrator we did not include an anatomical illustration in this manuscript. If Elsevier has rights to an anatomical illustration that we could use, we would be happy to include this in this manuscript.
  4. P3 Line 102: We addressed Reviewer 1's comments and suggestions
  5. Reviewer 1 indicates that text is duplicated on P 8 276-340 and P6 213-275. In reviewing the manuscript we found the text duplicated at a different location within the Discussion Section of the manuscript. The duplicated text was deleted. 

Reviewer 2 Report

This article deals with a novel ultrasound-guided approach combined with electromyography for botulinum toxin injection of longus colli and longus capitis. The technique proposed aims to improve the possibilities of a safe way of injection in the management of the patients with cervical dystonia due to the longus colli and longus capitis. The features of the technique are well described highlighting both the strengths and limitations.

As minor prompts:

  • could be preferable that keywords will be MeSH terms
  • materials and methods section should be move before discussion as point 3;
  • the discussion section is duplicate.

Author Response

Thank you to Reviewer 2 for your suggestions.

  1. We added the MeSH terms "dystonia" and "botulinum toxins" and also the entry term "Cervical Dystonia" which resides under torticollis. Anterocapitis and anterocaput and not MeSH terms but more accurately describe the condition in the manuscript.
  2. The Materials and Methods section is placed within the manuscript  after the discussion as required by the journal instructions to Authors
  3. The duplicated text was removed from the Discussion Section.
